# *FTO* rs9939609: T>A Variant and Physical Inactivity as Important Risk Factors for Class III Obesity: A Cross-Sectional Study

**DOI:** 10.3390/healthcare12070787

**Published:** 2024-04-04

**Authors:** Erika Martínez-López, Mariana Perez-Robles, Joel Torres-Vanegas, Sissi Godinez-Mora, Iris Monserrat Llamas-Covarrubias, Wendy Campos-Perez

**Affiliations:** 1Instituto de Nutrigenética y Nutrigenómica Traslacional, Departamento de Biología Molecular y Genómica, Centro Universitario de Ciencias de la Salud, Universidad de Guadalajara, Guadalajara 44340, Jalisco, Mexico; erika.martinez@academicos.udg.mx (E.M.-L.); mariana.perez@academicos.udg.mx (M.P.-R.); joel.torres3337@alumnos.udg.mx (J.T.-V.); sissi.godinez5582@alumnos.udg.mx (S.G.-M.); iris.llamas@academicos.udg.mx (I.M.L.-C.); 2Departamento de la Reproducción Humana, Crecimiento y Desarrollo Infantil, Centro Universitario de Ciencias de la Salud, Universidad de Guadalajara, Guadalajara 44340, Jalisco, Mexico

**Keywords:** class III obesity, physical inactivity, *FTO* rs9939609:T>A variant, BMI, waist circumference, nutrigenetics

## Abstract

Background: The prevalence of obesity has been increasing worldwide. It has been reported that physiological and environmental factors such as diet, culture, physical activity, and genetics are the principal factors related to obesity. The fat mass and obesity-associated (*FTO*) gen variant (rs9939609: T>A) has been associated with class III obesity. The A variant has been correlated with anthropometric and metabolic alterations. Therefore, the purpose of this study was to analyze the association of the *FTO* rs9939609: T>A variant and environmental factors with clinical, anthropometric, and biochemical variables in subjects with class III obesity. Results: The A variant frequency was higher in the class III obesity group compared with the normal weight group (44% vs. 25%, *p* < 0.001). Subjects with the AA genotype had a higher body mass index (BMI) than those with the AT genotype (35.46 kg/m^2^ (31–39.8) vs. 26.91 kg/m^2^ (23.7–30), *p* = 0.005). Women with the AA genotype showed higher waist circumferences than the AT group (101.07 cm (90.9–111.1) vs. 85.45 cm (77–93.8) *p* = 0.047). The *FTO* A variant increases the risk by 3.54 times and physical inactivity increases the risk by 6.37 times for class III obesity. Conclusions: Our results suggest that among the studied variables, those most related to class III obesity were the *FTO* risk genotype (A allele) and physical inactivity.

## 1. Introduction

The prevalence of obesity has been increasing worldwide. According to the ENSANUT survey (Encuesta Nacional de Salud y Nutrición), 36% of the Mexican population has obesity, of which 23.9% corresponds to class I obesity, according to the body mass index (BMI) proposed by the World Health Organization (WHO) (between 30 and 34.9 kg/m^2^—8.4% class II (35–39.9 kg/m^2^) and 3.7% class III (≥40 kg/m^2^)). It has been reported that environmental, physiological, dietetic, cultural factors, physical inactivity, and genetics have an important role in the development of the disease [1,2,3]. Moreover, BMI is an indicator of nutritional status, which exponentially increases the risk for chronic diseases like hypertension, dyslipidemia, diabetes mellitus type 2 (DM2), cardiovascular diseases (CVD), bile duct disease, sleep apnea, osteoarticular diseases, some types of cancer [4,5], and those with a solid psychological impact such as depression and anxiety [6]. Nevertheless, not all subjects respond the same to environmental factors related to obesity, probably due to genetic variants [7].

In recent years, 52 loci related to obesity have been identified by genome-wide association studies (GWAS) in which the fat mass and obesity-associated gene (*FTO*), also known as alpha-ketoglutarate-dependent dioxygenase, was one of the first obesity-associated loci described [8]. *FTO* is expressed ubiquitously with the highest expression in the liver, brain, visceral fat, and hypothalamus, and it is involved in the regulation of food intake and energy expenditure through the expression of gene modulation with an mRNA demethylase function [9,10]. The *FTO* rs9939609:T>A variant, located in intron 1, has been associated with obesity and increased risk of CVD [11,12].

Moreover, the A variant has been correlated with increased body weight, waist circumference (WC), BMI, body fat percentage (BF%), and blood pressure, as well as higher concentrations of total cholesterol (TC), low-density lipoprotein cholesterol (LDL-c), and triglycerides (TG), which exacerbate the risk for develop chronic diseases associated with obesity [13,14].

Therefore, the aim of this study was to analyze the association of the *FTO* rs9939609 variant and environmental factors with class III obesity and related variables, including those clinical, anthropometric, and biochemical in nature.

## 2. Materials and Methods

### 2.1. Study Population

In this cross-sectional study, a total of 169 subjects from West Mexico were recruited. A medical and nutritional screening by the Instituto de Nutrigenética y Nutrigenómica Traslacional, was performed on subjects who attended the institute; then, they were invited to participate in the study. They regularly attended nutrition assessments and counseling at the Universidad de Guadalajara Jalisco, México.

In both groups, the inclusion criteria were subjects 18 to 60 years old who signed the informed consent; for the controls, the criteria was a BMI of 18.5–24.9 kg/m^2^ with no diagnosed diseases nor medical or nutritional treatments, and for the class III obesity group, a BMI ≥ 40 kg/m^2^ was required, and they must have been scheduled for bypass surgery. Due to class III obesity being the last, most harmful stage of excess weight, these individuals were included to compare the variables with normal weight and find the most significant variables involved in obesity.

Subjects were excluded if they had incomplete data. Pregnant women or breastfeeding were not included.

The sample size was calculated using the OpenEpi version 3.01 software (https://www.openepi.com/Menu/OE_Menu.htm accessed on 18 July 2022) with Fleiss using a two-tail method with a confidence interval of 95% and a power of 80%. We considered the prevalence of the risk A allele in subjects with class III obesity to be 31.1%, as reported by Villalobos-Comparán et al. [15]; therefore, the minimum number of subjects required was 41 per group. The recruitment of the participants spanned from 2020 to 2022. Significant alcohol drinking was considered if alcohol intake was >14 g per day (one drink) and >28 g per day (two drinks) for women and men, respectively, in the past year [16]. Moreover, tobacco users were defined as those patients who had smoked at least one cigarette per day during the last 6 months [17]. 

### 2.2. Dietary Intake Assessment

Each subject was interviewed by a nutritionist. A dietary assessment of their habitual intake was performed using a habitual day food record (24 h recall) applied by the nutritionist, which was then analyzed in the Nutritionist Pro™ Diet analysis software version 8.1 (Axxya Systems, Stafford, TX, USA), designed for dietary evaluation. With the objective of obtaining more accurate information, food scales and models from Nasco^®^ (Fort Atkinson, WI, USA), as well as household measures (cups, spoons, among others), were used. Our methods aimed to estimate the actual quantity of food eaten, as previously described [18,19].

Simple carbohydrates (sCH) include all monosaccharides and disaccharides added by the manufacturer, cook, or consumer, plus sugars naturally present in honey, syrups, and fruit juices. The dietary polyunsaturated fatty acid (PUFA) n-6: n-3 ratio was calculated by dividing the n-6 value by the n-3 in grams.

### 2.3. Physical Activity 

If subjects engaged in at least 150 min per week of exercise at moderate intensity or at least 75 min of vigorous-intensity aerobic physical activity per week, or an equivalent combination of moderate- and vigorous-intensity activity for at least 4 weeks, they were considered to exercise in accordance with the WHO [20]. Intensity was measured based on the activities listed in the *Compendium of Physical Activities*, as previously described [18]. If subjects reported not performing these activities or did not perform them for the times outlined above, they were classified as physically inactive.

### 2.4. Biochemical Analysis

Blood samples were taken after 8 to 10 h of fasting; then, they were centrifuged at 4 °C for 15 min at 3500 r/min to obtain the serum. Serum glucose, TG, TC, and high-density lipoprotein cholesterol (HDL-c) were determined by dry chemistry using a Vitros 350 Analyzer (Ortho-Clinical Diagnostics, Johnson and Johnson Services Inc., Rochester, NY, USA). LDL-c was calculated using the Friedewald formula as long as TG levels were less than 400 mg/mL [21]. The TG/HDL-c and TC/HDL-c ratios were calculated by dividing the TG by HDL-c and TC by HDL-c, in mg/dL, respectively.

### 2.5. Anthropometric and Clinical Measurements 

Measurements were performed after 8 to 10 h of fasting. Participants wore light clothes and were barefoot. Height was determined using a stadiometer with a precision of 1 mm (Rochester Clinical Research, Inc., New York, NY, USA). WC was measured according to the ISAK method using a Lufkin Executive^®^ (New Brighton, MN, USA) thinline 2 mm measuring tape. Tetrapolar body electrical bioimpedance was used to assess body fat percentage (BF%) (InBody 370, Biospace Co., Seoul, Republic of Korea). BMI was calculated as weight in kilograms divided by height in meters squared (kg/m^2^).

Blood pressure was measured with a pulse sphygmomanometer (Omron). Once the dominant arm was identified, the patient relaxed in a seated posture for 15 min, with the arm supported at heart level, the back supported, and the feet resting freely on the floor without crossing the legs. The mean of two measurements was reported.

### 2.6. DNA Extraction and Genotyping 

Genomic DNA (gDNA) was extracted from peripheral leukocytes using the High Pure PCR Template Preparation Kit (Roche Diagnostics, Indianapolis, IN, USA). The *FTO* variant (rs9939609) was determined by allelic discrimination using TaqMan^®^ probes, catalog number 4351379 (Drug Metabolism Assay; Applied Biosystems, Foster City, CA, USA). The final gDNA concentration was 20 ng/µL. The experiments were carried out in a LightCycler^®^ 96 RealTime Polymerase Chain Reaction System (Roche Diagnostics, Mannheim, Germany) under the following conditions: 95 °C for 10 min and 40 cycles of denaturation at 95 °C for 15 s and annealing/extension at 60 °C for 1 min. The *FTO* rs9939609 genotyping was verified using positive controls of the gDNA samples to the three possible genotypes in each 96-well plate.

### 2.7. Statistical Analysis

To analyze the distribution of variables, the Shapiro–Wilk test was used. Descriptive quantitative variables were expressed as medians and interquartile ranges. Categorical variables were expressed as numbers and percentages. For the comparison of quantitative variables, the univariate general linear model adjusted by intervening variables (age, sex, alcohol consumption, physical activity, and energy intake) was used. These data were reported in estimated means and confidence intervals. Moreover, the chi-square test was used to compare the qualitative variables and the genotypic and allelic frequencies in the study groups, followed by a Bonferroni test. Finally, using a binary logistic regression model, the variables that could be related to a higher risk for class III obesity were analyzed. Statistical analyses were performed using SPSS version 20.0 software (IBM Corp., Armonk, NY, USA), and a *p*-value < 0.05 was considered statistically significant.

### 2.8. Statement of Ethics

This study was approved by the Ethics Research Committee “Comité de investigación, ética en investigación y de Bioseguridad” of the Centro Universitario de Ciencias de la Salud, Guadalajara, Jalisco, Mexico (number: CI-08320). Subjects who agreed to participate were informed about the research procedures and signed the informed consent. All procedures were conducted according to the Declaration of Helsinki (2013) guidelines and the American Medical Association (2013), as well as what is stipulated in the Regulation of the General Law of Health in the Matter of Research for Health (2014), considering ethical aspects that guarantee the dignity and well-being of the person subject to research.

Moreover, the International Ethical Guidelines for Biomedical Research Involving Human Beings established by the Council for International Organizations of Medical Sciences (CIOMS) were followed, specifically guideline number 12, which deals with the collection, storage, and use of data in research related to health.

## 3. Results

### 3.1. Population Description

In this study, a total of 169 subjects were included (Figure 1); 20.7% were men (*n* = 35) and 79.3% were women (*n* = 134). 

The median age was 34 (27–43) years old. The quantitative and qualitative variables are described in Table 1. 

When comparing the observed vs. expected genotype frequencies, no statistically significant differences were found (*p* = 0.150), therefore, it was assumed that the population is in Hardy-Weinberg equilibrium.

### 3.2. Characteristics of the Study Groups

With respect to age, a median of 31.5 (24–43) years was obtained in subjects with normal weight vs. 38 (30–44) years in subjects with class III obesity (*p* = 0.014).

Regarding the clinical variables, significantly higher values of systolic blood pressure (SBP) (122.26 mmHg (117.6–126.8) vs. 112.18 mmHg (109.1–114.8), *p* < 0.001) and diastolic blood pressure (DBP) (80.10 mmHg (77–83.1) vs. 70.28 mmHg (68.4–72.1) *p* < 0.001) were found in the class III obesity vs. the normal weight group, as well as higher serum levels of TG (154.86 mg/dL (131.2–178.5) vs. 121.41 mg/dL (107.4–135.3) *p* = 0.021), TG/HDL-c ratio (4.94 (4.2–5.6) vs. 2.77 (2.3–3.1) and TC/HDL-c ratios (5.54 (5.1–5.9) vs. 3.94 (3.7–4.1) *p* < 0.001). Moreover, a lower HDL-c was found in women with class III obesity compared with those of a normal weight (33.76 mg/dL (29.8–37.6) vs. 50.51 mg/dL (48–52.9) *p* < 0.001) (Table 1). 

When comparing the dietary intake between study groups, it was found that subjects with class III obesity have a significantly higher intake of total fatty acids (TFAs) (38.49% (36.1–40.8) vs. 34.89% (33–36.7) *p* = 0.020) and PUFAs (9.15% (7.8–10.4) vs. 5.21% (4.1–6.2) *p* < 0.001); however, they also have a higher n-6:n-3 ratio (18.55:1 g (15.9–21.1) vs. 11.28:1 g (9.2–13.2) *p* < 0.001) as well as lower fiber intake (17.25 g (14.6–19.8) vs. 27.26 g (23.9–30.5) *p* < 0.001). On the other hand, subjects with normal weight showed a higher intake of simple carbohydrates (sCH) than subjects with class III obesity (11.68% (9.9–13.3) vs. 8.17% (5.4–10.8) *p* = 0.033) (Table 1).

Regarding the qualitative variables, a greater frequency of physical inactivity was found in subjects with class III obesity (82% vs. 44%, *p* < 0.001). Nonetheless, a higher frequency of drinking alcohol was found in subjects with normal weight (47% vs. 22%, *p* = 0.002) (Table 1).

#### FTO Genotype and Allelic Frequencies

The genotype frequencies were compared in the study groups (Table 1), and a significantly higher frequency of the AA genotype was found in subjects with class III obesity compared with the normal weight group (35% vs. 12%, *p* = 0.025). Likewise, a higher frequency of the A allele was found in the group of patients with class III obesity than in the group of subjects with normal weight (44% vs. 25%, *p* < 0.001).

### 3.3. Characteristics of Study Population According to FTO rs9939609 Variant 

When comparing the study variables in the entire population classified by the different *FTO* genotypes, it was observed that subjects with the AA genotype had a higher BMI than those with the AT genotype (35.46 kg/m^2^ (31–39.8) vs. 26.91 kg/m^2^ (23.7–30), *p* = 0.005). Likewise, women with the AA genotype showed a higher WC than carriers of the AT genotype (101.07 cm (90.9–111.1) vs. 85.45 cm (77–93.8) *p* = 0.047) (Table 2).

### 3.4. Risk Factors for Class III Obesity 

Using a binary logistic regression model, the variables that could be related to higher risk for class III obesity were analyzed, including all dietary and qualitative variables described above. The *FTO* rs9939609 variant increases the risk for class III obesity 3.54 times, and physical inactivity increases the risk 6.37 times (Table 3).

## 4. Discussion

It is well known that environmental factors including inadequate diet, physical inactivity, and genetics are related to obesity; however, it is important to identify the most relevant factors in each population. In the present study, it was found that the *FTO* rs9939609 variant and physical inactivity are the main variables related to an increased risk for class III obesity. 

Obesity has been rising in epidemic proportions and has become a serious health problem that contributes to the development of numerous metabolic comorbidities [22]. It has been described that people with class III obesity have impaired lipid metabolism, characterized by elevated levels of serum TG, VLDL-c, and LDL-c, and low HDL-c [23]. In this sense, González-Sánchez et al., found, in 732 subjects from Spain, higher TG in subjects with obesity compared with those without (141.5 ± 53 mg/dL vs. 106.1 ± 44.2 mg/dL, *p* < 0.001), lower HDL-c levels (50.1 mg/dL ± 15.4 mg/dL vs. 57.9 ± 19.3 mg/dL, *p* < 0.001), higher TC (223.9 ± 46.3 mg/dL vs. 208.4 ± 38.61 mg/dL, *p* < 0.001), and higher LDL-c (142.8 ± 34.7 mg/dL vs. 131.2 ± 30.8 mg/dL, respectively, *p* < 0.001) [24].

Related to inadequate environmental factors, an unbalanced diet was found not only in subjects with excess weight but also in normal-weight subjects [25]. These data are consistent with a previous study by our research group, carried out in 500 Mexicans, where it was found that the group of patients with and without obesity had an excessive intake of CHs, TFAs, and SFAs and an inadequate PUFA n-6:n-3 ratio [26]. Moreover, a higher prevalence of alcohol consumption was found in normal-weight subjects; therefore, if they do not modify their lifestyle features, this could contribute to the development of obesity and its comorbidities.

Another important environmental factor is physical activity because it reduces adipose tissue and promotes muscle gain while improving metabolism [27,28]. Regarding this, we found a higher frequency of physical inactivity in subjects with class III obesity. Like these, Kim et al. [29] found that participants who did not engage regularly in physical activity exhibited a higher BMI than those who were physically active (30.1  ±  7.3 kg/m^2^ vs. 27.9 ± 6 3 kg/m^2^, respectively, *p* < 0.001); however, we detected that physical active subjects were younger and, therefore, age might play an important role in obesity.

Regarding genetic factors, the *FTO* rs9939609:T>A variant is one of the most studied [30,31]. The A allele frequencies found in this study were higher than in other studies previously carried out in the population of America (26%) and subjects with Mexican ancestry living in Los Angeles (22%) [32]. These differences could be attributed to the fact that patients with class III obesity were included in this study as a group, and the A variant is considered a risk factor [15].

In addition, the A variant and AA genotype were higher in subjects with class III obesity than the group of subjects with normal weight. González-Sánchez et al. obtained similar results. They found that the AA genotype was significantly more present in individuals with obesity than in those without (19.9% vs. 13.7%, *p* = 0.026) [24]. Likewise, in a study carried out on the Mexican population, Villalobos-Comparán et al. [15] reported a higher frequency of the A variant in subjects with class III obesity (31%) vs. class I/II obesity and subjects living without obesity (21% and 17%, respectively). Furthermore, they also found that the AA genotype was significantly more frequent in subjects with class III obesity (10.8%) than class I/II obesity and subjects living without obesity (5.6% and 3.1%, respectively).

Interestingly, when comparing the different *FTO* genotypes, it was found that subjects with the AA genotype had higher BMI than those with the AT genotype. Likewise, women with the AA genotype showed a higher WC, which is related to abdominal obesity and linked to higher CVD risk factors like hypertension and dyslipidemia [33]. In a study with Brazilian youths, a strong association of variant A with higher BMIs and WCs was found (OR 3.21, 95% CI 1.71–6.05, *p* < 0.001; OR 2.59, 95% CI 1.35–4.97, *p* = 0.004, respectively) [34]. In agreement, in a study carried out on Finnish subjects, those with the AA genotype had higher BMIs (32.6 ± 4.7 kg/m^2^, vs. TT: 30.7 ± 4.4 kg/m^2^ and TA: 31.1 ± 4.4, *p* = 0.004) [35]; however, as a result of only having one subject with the AA genotype, more studies are needed to corroborate our results. Furthermore, according to our results, in a Danish population with DM2, Andreasen et al. [36] reported greater BMIs in AA carriers who were physically inactive vs. TT carriers who also were physically inactive (27 ± 4.9 vs. 25.9 ± 7.9 kg/m^2^, *p* < 0.001), which also highlights the strong effect that gene–environment interaction has. Therefore, it is important to promote physical activity, especially in subjects with risk genotypes.

These results highlight the importance of promoting a healthy lifestyle because most of the risk factors for obesity could be modifiable. Thus, by reducing or eliminating exposure to these risk factors, the probability of obesity decreases. However, it is necessary to consider the genetic variants in each population that most predispose to obesity and comorbidities to offer better treatments following the principles of nutrigenetics and precision medicine. A limitation of this study is the small number of patients with class III obesity and lacking enough males to compare all variables properly; however, for future research, the sample size would be expanded as it has been reported that gender differences are present in the associations of the *FTO* rs9939609 with obesity-related traits [37]. In future studies, wearable technology is recommended to measure physical activity more accurately. Moreover, another avenue involves detecting *FTO* mRNA expression in adipose tissue, offering valuable data for basic research.

## 5. Conclusions

Our results suggest that among the variables studied, the most associated with risk for class III obesity were the *FTO* rs9939609 AA risk genotype and physical inactivity. Therefore, it is important to promote more physical activity, especially in subjects with the risk genotype. 

## Figures and Tables

**Figure 1 healthcare-12-00787-f001:**
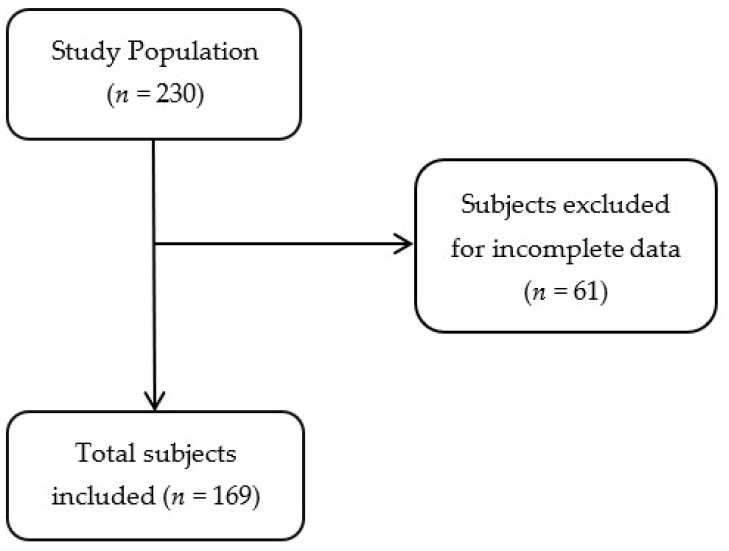
Flow diagram of study subjects.

**Table 1 healthcare-12-00787-t001:** Characteristics of the study subjects.

Variables	General Population (*n* = 169)	Class III Obesity (*n* = 55)	Normal Weight (*n* = 114)	*p* Value
**Sociodemographic and anthropometrical**				
Sex M/W (%)	20.7/79.3	14.5/85.5	23.7/76.3	0.170
Age (years)	34 (27–43)	38 (30–44)	31.5 (24–43)	0.014
SBP (mm Hg) ^a^	115 (103–126)	122.26 (117.6–126.8)	112.18 (109.1–114.8)	<0.001
DBP (mm Hg) ^a^	73 (65–80)	80.10 (77–83.1)	70.28 (68.4–72.1)	<0.001
Weight (kg) ^a^	61 (55–68.5)	117.78 (113.18–122.38)	60.35 (58.81–61.89)	<0.001
BMI (kg/m^2^) ^a^	23.85 (21.52–40.1)	43.69 (42.57–44.81)	22.32 (21.65–22.99)	<0.001
Body Fat (%) ^a^				
M (*n* = 35)	21 (18–23.5)	43.54 (36.71–50.38)	19.27 (17.54–21)	<0.001
W (*n* = 134)	30 (24.25–33)	48.96 (45.15–52.76)	28 (26.71–29.29)	<0.001
WC (cm) ^a^				
M (*n* = 35)	88 (82–101)	140.33 (133.66–147)	86.05 (83.21–88.88)	<0.001
W (*n* = 134)	79.5 (71.7–114.2)	118.48 (115.49–121.47)	75.12 (73.18–77.06)	<0.001
**Biochemical**				
TC (mg/dL) ^a^	175 (155–205)	177.9 (165.9–189.9)	183.07(176–190.1)	0.483
LDL-c (mg/dL) ^a^	103.5 (88.2–129)	113.11 (103.1–123)	110.06 (104–116)	0.618
HDL-c (mg/dL) ^a^				
M (*n* = 35)	38 (34–45)	34.18 (24.1–44.1)	43.24 (38.9–47.5)	0.104
W (*n* = 134)	44 (35–54)	33.76 (29.8–37.6)	50.51 (48–52.9)	<0.001
TG (mg/dL) ^a^	116 (81.7–148.2)	154.86 (131.2–178.5)	121.41 (107.4–135.3)	0.021
TG/HDL-c ratio ^a^	2.7 (1.61–4.48)	4.94 (4.2–5.6)	2.77 (2.3–3.1)	<0.001
TC/HDL-c ratio ^a^	4.15 (3.27–5.45)	5.54 (5.1–5.9)	3.94 (3.7–4.1)	<0.001
**Dietary intake**				
Energy Intake (kcal) ^b^	2070.5 (1723.8–2576.3)	2147.4 (1904.1–2390.6)	2211.4 (2013.2–2409.5)	0.688
CH (%) ^b^	48.79 (41.39–52)	46.38 (43.9–48.8)	48.18 (46.1–50.1)	0.267
sCH (%) ^b^	11.1 (6.2–16)	8.17 (5.4–10.8)	11.68 (9.9–13.3)	0.033
Proteins (%) ^b^	15.77 (13.6–18.49)	16.41 (15.1–17.6)	16.45 (15.4–17.4)	0.959
TFAs (%) ^b^	36.79 (30–39.69)	38.49 (36.1–40.8)	34.89 (33–36.7)	0.020
SFAs (%) ^b^	10.51 (8.17–13)	11.02 (10–11.9)	10.65 (9.8–11.4)	0.552
MUFAs (%) ^b^	10.92 (8.36–13)	11.32 (10–12.6)	10.82 (9.78–11.8)	0.562
PUFAs (%) ^b^	6 (4–8)	9.15 (7.8–10.4)	5.21 (4.1–6.2)	<0.001
n-6:n-3 ratio ^b^	10.69 (8.25–17.85)	18.55 (15.9–21.1)	11.28 (9.2–13.2)	<0.001
Cholesterol (mg) ^b^	242.33 (153–328.3)	290.88 (230.7–351)	268.94 (222–315.7)	0.572
Fiber (g) ^b^	17.93 (13.1–26.4)	17.25 (14.6–19.8)	27.26 (23.9–30.5)	<0.001
**Genotype frequency**				
TT, *n* (%)	95 (56)	26 (47)	69 (60)	0.025
AT, *n* (%)	42 (25)	10 (18)	32 (28)	
AA, *n* (%)	32 (19)	19 (35) ^c^	13 (12) ^c^	
**Allelic distribution**				
A, *n* (%)	52 (31)	24 (44)	28 (25)	<0.001
T, *n* (%)	117 (69)	31 (56)	86 (75)	
**Qualitative variables**				
Physical inactivity, *n* (%)	95 (56)	45 (82)	50 (44)	<0.001
Tobacco users, *n* (%)	41 (24)	15 (28)	26 (23)	0.105
Alcohol drinkers, *n* (%)	65 (38)	12 (22)	53 (47)	0.002

^a^ Data reported in estimated mean and confidence intervals, adjusted for age, sex, alcohol consumption, and physical activity. ^b^ Data also adjusted by total kcal. ^c^ Differences between groups. The *p*-value was obtained by the differences between the study groups using the univariate general linear model. SBP: Systolic blood pressure, DBP: Diastolic blood pressure, BMI: Body mass index, M: Men, W: Women, WC: Waist circumference, TC: Total serum cholesterol, LDL-c: Low-density lipoprotein cholesterol, HDL-c: High-density lipoprotein cholesterol. TG: Triglycerides, CH: Carbohydrates, sCH: Simple carbohydrates, TFAs: Total fatty acids, SFAs: Saturated fatty acids, MUFAs: Monounsaturated fatty acids, PUFAs: Polyunsaturated fatty acids, n-6: omega 6, n-3: omega 3.

**Table 2 healthcare-12-00787-t002:** Quantitative variables in subjects classified by genotypes of the *FTO* rs9939609 variant.

Variables	TT (*n* = 95)	AT (*n* = 42)	AA (*n* = 32)	*p* Value
Sex (M/W), %	20/80	28.6/71.4	3.1/96.9	0.175
Age (years)	35.80 (33.48–38.12)	35.11 (31.60–38.63)	33.35 (28.48–38.22)	0.616
**Clinical**				
SBP (mm Hg)	114.38 (111.5–117.2)	114.19 (109.8–118.5)	119.12 (112.8–125.4)	0.376
DBP (mm Hg)	72.86 (69.7–75.9)	72.86 (69.7–75.9)	75.08 (70.5–79.6)	0.487
**Anthropometrical**				
Weight (kg)	64.08 (59.4–68.7)	68.3 (61.6–75)	75.5 (64–86.9)	0.151
BMI (kg/m^2^)	28.02 (25.9–30.1)	26.91 (23.7–30) ^a^	35.46 (31–39.8) ^a^	0.005
Body Fat (%)				
M (*n* = 35)	19.59 (15.6–23.5)	23.21 (18.6–27.7)	24.12	0.610
W (*n* = 134)	29.87 (27.3–32.3)	29.62 (25.8–33.4)	32.98 (27.6–38.2)	0.540
WC (cm)				
M (*n* = 35)	93.87 (83.4–104.2)	94.91 (81.8–108)	144.35	0.104
W (*n* = 134)	88.21 (82.9–93.5)	85.45 (77–93.8) ^a^	101.07 (90.9–111.1) ^a^	0.047
**Biochemical**				
TC (mg/dL)	178.17 (171.1–185.1)	184.08 (173.5–194.5)	176.44 (161–191.3)	0.596
LDL-c(mg/dL)	107.62 (101.7–113.4)	112.49 (103.5–121.4)	110.73 (98.4–123)	0.647
HDL-c(mg/dL)				
M (*n* = 35)	40.03 (35.1–44.8)	45.08 (38.9–51.1)	30.39	0.246
W (*n* = 134)	45.51 (42.3–48.7)	47.99 (42.9–53)	41.97 (35.8–48)	0.328
TG (mg/dL)	127.39 (113–141.7)	122.09 (100.6–143.5)	125.23 (94.6–155.7)	0.920
TG/HDL-c ratio	3.28 (2.8–3.7)	2.98 (2.2–3.7)	3.63 (2.6–4.6)	0.574
TC/HDL-c ratio	4.31 (4–4.5)	4.12 (3.7–4.5)	4.69 (4.09–5.3)	0.318

Data reported in estimated mean and confidence intervals, adjusted by the variables age and sex. The *p*-value was obtained using the univariate general linear model. SBP: Systolic blood pressure, DBP: Diastolic blood pressure, M: Men, W: Women, TC: Total serum cholesterol, LDL-c: Low-density lipoprotein cholesterol, HDL-c: high-density lipoprotein cholesterol. TG: triglycerides. ^a^ Differences between groups.

**Table 3 healthcare-12-00787-t003:** Risk factors associated with class III obesity.

Variables	B	*p* Value	OR	95% CI
*FTO* rs9939609:T>A variant	1.265	0.002	3.544	1.347–9.322
Physical inactivity	1.853	<0.001	6.377	2.55–5.914

Logistic binary regression test adjusted by age and sex.

## Data Availability

The datasets used and/or analyzed during the current study are available from the corresponding author upon reasonable request.

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
