# Peer review of "FTO rs9939609: T>A Variant and Physical Inactivity as Important Risk Factors for Class III Obesity: A Cross-Sectional Study"

_healthcare, 2024, doi:10.3390/healthcare12070787_

Round 1

Reviewer 1 Report (Previous Reviewer 3)

Comments and Suggestions for Authors

This manuscript is the resubmission of healthcare-2852311. The authors responded to and revised their manuscript following the reviewer’s comments. Now, the reviewer thinks it deserves publication.

Author Response

Thank you very much for your time, comments, and review.

Reviewer 2 Report (Previous Reviewer 2)

Comments and Suggestions for Authors

Thank you for addressing all comments.

Author Response

Thank you very much for your review and comments to improve the article.

Reviewer 3 Report (Previous Reviewer 1)

Comments and Suggestions for Authors

Still the number of recruited subjects is low to drag any conclusions.

Author Response

Thank you for your time and observations. The small sample size has been added as a limitation for the study. We are working on expanding the number of subjects to have better statistical results in future papers (lines 275-276, 287-290). 

This manuscript is a resubmission of an earlier submission. The following is a list of the peer review reports and author responses from that submission.

Round 1

Reviewer 1 Report

Comments and Suggestions for Authors

The authors in this study analyzed the association of the FTO rs9939609: T>A variant and environmental factors with clinical, anthropometric, and biochemical variables in subjects with class III obesity. They found that A variant frequency was higher in class III obesity group compared with normal weight group (p<0.001). the conclusions of the results is that the most related with class III obesity were the FTO risk genotype (A allele) and physical inactivity.

- Although the study is interesting and novel, the main concern is the limited number of recruited patients in the investigation (169), which is  very low to draw a conclusion.

Reviewer 2 Report

Comments and Suggestions for Authors

The research investigates the association between the FTO rs9939609 variant, physical inactivity, and class III obesity in a Mexican cohort. It finds that the A allele of the FTO variant and physical inactivity significantly correlate with class III obesity, highlighting the genetic and environmental factors' roles in obesity. The study emphasizes the importance of physical activity, especially for those with the risk genotype, to mitigate obesity risks. The study provides valuable insights into the genetic and lifestyle factors contributing to class III obesity. The research is well-designed, with clearly presented results. The discussion on the findings is insightful. However, In my opinion, Several questions need addressing for the acceptance of this manuscript.

1. A significant limitation of this study is the small sample size for class III obesity participants. The researcher should consider to increase the size of sample. Besides increasing the sample size, the author also can consider to use wearable technology to gather real-time, objective data on diet intake and physical activity.

2. Why are some indexes, like body fat and WC, compared separately for women and men in Table 1, while others, like weight and BMI, are compared overall? Consistency is essential in all comparisons.

3. The author exclusively examined serum levels of TG, LDL, and HDL in participants. However, these biomarkers alone may not sufficiently characterize individuals' metabolic status. Measurements of fasting insulin, glucose levels, and HOMA-index should also be conducted.

4. In the study, the author genotyped the FTO variant in peripheral leukocytes of participants. However, understanding how the FTO variant affects FTO gene expression in metabolically active tissues, like subcutaneous adipose tissue, is crucial. The author should consider detecting FTO mRNA expression in adipose tissue, which would offer valuable insights for basic research.

5. In the results section 3.2, line 191-194, it was stated that “Regarding the qualitative variables, a greater frequency of physical inactivity was found in subjects with class III obesity (83% vs 43%, p<0.001). Nonetheless, a higher frequency of drinking alcohol was found in subjects with normal weight (47% vs 22%, p=0.002) (Table 1). ” However, the data presented in Table 1 is opposite. Which one is correct?

6. In the results section 3.3, it's noted that women with the AA genotype have a higher waist circumference (WC) than those with the AT genotype, but no difference is observed among men. Does this imply that there are gender differences in FTO variant associated with obese index?

7. The manuscript contains minor errors, such as the incorrect formatting of "kg/m2" on lines 37, 38, 68, 213.., where "2" should be superscript. 

Reviewer 3 Report

Comments and Suggestions for Authors

The authors investigated the association of the FTO rs9939609: T>A variants and environmental factors with clinical, anthropometric, and biochemical variables in subjects with class III obesity. The FTO A variant increases the risk for class III obesity 3.54 times, and physical inactivity increases the risk 6.37 times.

The experiments and analyses seemed to be performed properly, and the results were clear and straightforward. However, the reviewer has some concerns to clarify for publication consideration, as described below.

1.       Why did the authors focus on only class III obesity? Class I and II obesities also impact the risk for non-communicable diseases. FTO gene variants are not specific to class III obesity.

2.       In lines 133-134 in the Materials and Methods section, they wrote that data were expressed as mean ± SD. However, most of the data in Tables 1 and 2 were probably expressed as median (10 percentile – 90 percentile). Please define concretely the way of description. Furthermore, it would be better to unify the expression throughout the manuscript.

3.       In the Qualitative variables in Table 1, the percentage description seemed wrong. For example, the number of the person with physical inactivity of class III obesity was 45. Total number of class III was 55. Therefore, the rate should be 82%

4.       In the Quantitative variables of Table 2, the reviewer thinks that the number of AA male variants should be one because the total number of participants with AA variants was 24, and the rate of the male was 4.5% (24 multiplies with 0.045 is 1.08). Then, why did the Body fat, WC, and HDL-c of male subjects with AA have variables?

5.       Why did the authors combine male and female subjects for the analysis? The association of the A variants with obesity might be female-specific. Please check the paper with doi: 10.1016/j.bbrc.2008.01.087.

6.       The definition of the obesity class should be included in lines 33-34.

7.       Statistics should be performed to compare normal and obesity levels. Therefore, it would be better to use Welch’s t-test (parametric) or Mann-Whitney’s U-test (non-parametric).

8.       Clarify the statistical methods used in the Tables when P-values are shown.